# Temporal Feature Extractors in EEG Foundation Models: A Controlled Comparison Including a Pretrained Time-Series Model

Ayşe Betül Yüce [1]  Chris Joey Leffler [1]  Sarun Varghese [2]  Myra Spiliopoulou [2]  Sebastian Stober [1]

## Abstract

Electroencephalography (EEG) foundation models aim to learn generalizable representations from large-scale brain recordings. However, the role of temporal feature extractors and whether pretrained time-series foundation models (TSFMs) can be effectively transferred to this setting remains underexplored. We conduct a controlled comparison of three temporal feature extraction strategies, including a linear baseline, a convolutional encoder, and a frozen pretrained TSFM (MOMENT), within a unified EEG foundation model. We evaluate their impact on representation quality using two downstream tasks: motor imagery and emotion recognition. Results reveal different trends across the evaluated benchmarks. On the motor imagery dataset, simple temporal representations perform competitively, whereas the emotion dataset benefits from richer temporal modeling. Although not specifically adapted to EEG, the pretrained TSFM serves as an effective temporal feature extractor, suggesting that general-purpose time-series representations can be transferred as frozen temporal feature extractors within EEG foundation models.

## 1. Introduction

EEG provides noninvasive, high temporal resolution insights into brain activity with applications in clinical diagnosis and brain-computer interfaces. Recently, EEG foundation models have emerged to learn robust representations that generalize across subjects, datasets and tasks (Lai et al., 2025; Klein et al., 2026).

However, EEG data is challenging due to high dimensionality, non-stationarity, low signal-to-noise ratio and substantial inter-subject and task variability (Lai et al., 2018; Tran et al., 2026). With a significant portion of information encoded in temporal dynamics, temporal feature extraction is a crucial step for representation learning.

EEG foundation models employ an embedding module to transform raw EEG data into embeddings that are subsequently refined by a transformer backbone. A common approach is the use of lightweight 1D convolutional encoders (Kostas et al., 2021; Jiang et al., 2024), often combined with frequency-domain features (Wang et al., 2025; Döner et al., 2025), while some other architectures rely on linear projection layers only (Wang et al., 2024; Ouahidi et al., 2025; Klein et al., 2025). As differences in the embedding module are often entangled with variations in overall architecture and training paradigms, the direct effect of temporal feature extraction has not been systematically studied. This motivates investigating whether richer temporal representations could improve generalization across subjects and tasks.

Recent time-series foundation models (TSFMs) offer a potential alternative by learning general-purpose temporal representations from large, diverse datasets (Kottapalli et al., 2025). However, it remains unclear whether such representations transfer effectively to EEG, and can serve as plug-in temporal feature extractors in EEG foundation models.

In this study, we conduct a controlled comparison of temporal feature extractors in EEG foundation models under a unified setup. Specifically, we compare a linear projection, a depthwise separable convolutional encoder, and a frozen pretrained TSFM (MOMENT) (Goswami et al., 2024). We evaluate the quality of learned representations on two downstream tasks. The contributions of this work are as follows:

- We systematically analyze the role of temporal feature extractors in EEG foundation models across two downstream tasks.

- We evaluate a pretrained TSFM (MOMENT) as a frozen temporal feature extractor within an EEG foundation model through a controlled comparison with linear and convolutional embedding strategies.

---

[1]Otto von Guericke University Magdeburg, Artificial Intelligence Lab, Faculty of Computer Science, Magdeburg, Germany [2]Otto von Guericke University Magdeburg, Knowledge Management and Discovery Lab, Faculty of Computer Science, Magdeburg, Germany. Correspondence to: Ayşe Betül Yüce <ayse.yuece@ovgu.de>.

*Proceedings of the 2nd ICML Workshop on Foundation Models for Structured Data*, Seoul, South Korea. 2026. Copyright 2026 by the author(s).

## 2. Related Work

**EEG Foundation Models.** The first EEG foundation model, BENDR (Kostas et al., 2021), uses a 1D convolutional encoder to embed raw EEG signals, pretraining via masked modeling in latent space. Similarly, LaBraM (Jiang et al., 2024) adopts a patch-based strategy, where convolutional embeddings are mapped to discrete neural tokens using vector quantization. The model is then pretrained through direct token prediction. CBraMod (Wang et al., 2025) introduces parallel temporal and frequency-domain branches in the embedding module, an approach followed by LUNA (Döner et al., 2025). EEGConformer (Song et al., 2023) further extends the convolutional encoder with spatial convolutions over channels and self-attention layers. In contrast, EEGPT (Wang et al., 2024) and REVE (Ouahidi et al., 2025) remove convolutional components entirely and instead rely on linear projection layers.

**Time-Series Foundation Models (TSFMs).** TSFMs aim to learn general-purpose representations of temporal data from large and diverse pretraining corpora. Models such as Chronos (Ansari et al., 2024) and TimesFM (Das et al., 2024) are primarily designed for univariate forecasting, employing encoder-decoder and decoder-only transformer architectures, respectively. Moirai (Woo et al., 2024) expands this paradigm to multivariate forecasting tasks. While these models demonstrate strong zero-shot forecasting performance on unseen datasets, their training objectives are optimized for predictive modeling which makes their suitability as general-purpose feature extractors less clear.

In contrast, MOMENT (Goswami et al., 2024) uses a patch-based masked reconstruction objective designed for representation learning. It uses a large scale encoder-only transformer pretrained on diverse datasets spanning multiple domains including economics, weather, healthcare, and EEG sensor data. Its representation-focused training objective and downstream classification performance motivate its use as a frozen temporal feature extractor in this study.

## 3. Methodology

We study the role of temporal feature extraction in EEG foundation models under a controlled setup (Fig. 1), where all components are held fixed except the extractor. We compare three strategies: a linear projection, a convolutional encoder, and a frozen pretrained TSFM (MOMENT), and assess whether domain-general time-series representations transfer effectively to EEG.

### 3.1. EEG Input Representation and Patching

The input is a multichannel EEG recording represented as $X \in \mathbb{R}^{C \times T}$, where $C$ is the number of channels and $T$ denotes the number of time points. Following common practice in EEG foundation models, we extract patch representations as in Vision Transformers (Dosovitskiy et al., 2020). We divide each recording into 2-second patches and patched EEG is represented as $x \in \mathbb{R}^{C \times w \times t}$, where $t$ denotes the number of samples per patch and $w = \frac{T}{t}$ the number of patches per channel.

**Normalization.** We employ different normalization strategies depending on the temporal feature extractor. For the pretrained MOMENT-based setup, we rely on its native preprocessing, which includes reversible instance normalization (RevIN) (Kim et al., 2022), and do not apply additional normalization. For the other experiments, we employ absolute maximum normalization as an EEG-specific preprocessing choice, defined as $\tilde{X}_{c,t} = \frac{X_{c,t}}{\max(\max_t |X_{c,t}|, \epsilon)}$ where $\epsilon$ prevents division by zero. This normalization preserves both zero-crossings and signal polarity, which are important characteristics of EEG data.

### 3.2. Temporal Feature Extractor Strategies

We investigate the impact of the temporal feature extractor by comparing three approaches. All experiments share the same input representation, patching strategy, masking procedure, and transformer backbone. The only differences lie in the temporal feature extraction pipeline and the associated signal normalization.

**Linear Projection.** As a minimal baseline, each EEG patch is projected into the model embedding space with a linear layer.

**Convolutional Temporal Encoder.** Consistent with prior EEG foundation models (Wang et al., 2025; Jiang et al., 2024), we employ a convolutional encoder that applies shared temporal filters to each channel-patch token independently. The input is rearranged into separate temporal segments, and convolutions operate only along the temporal dimension using kernels of different sizes. Depthwise temporal convolutions are followed by a pointwise convolution to combine feature maps, after which the features are flattened and projected to the model embedding dimension.

**Pretrained TSFM (MOMENT).** We use the pretrained MOMENT-small model as a temporal feature extractor. This variant is selected due to computational constraints and its embedding dimensionality, which can be aligned with the model embedding space. Unlike forecasting-oriented TSFMs, MOMENT is pretrained with a masked reconstruction objective designed for representation learning, making it a suitable candidate for transfer to EEG.

The input is rearranged so that each channel-patch token is processed independently. Each input patch contains 200 time points, whereas MOMENT is originally trained with sequences of length 512. To handle this mismatch, we follow the model's native approach by applying left zero-

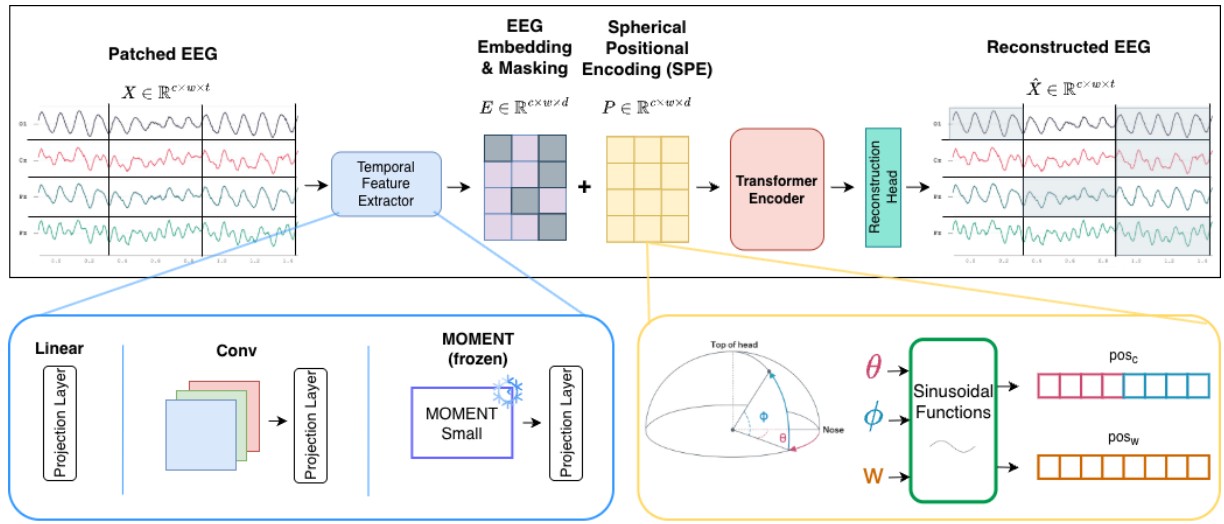

*Figure 1.* **Overview of the pretraining EEG foundation model**. The top block presents the pretraining pipeline. The bottom-left (blue) block illustrates the temporal feature extractor variants, and the bottom-right (yellow) block shows the Spherical Positional Encoding.

padding and providing a corresponding input mask. We use mean reduction over the internal patch representations of MOMENT, where internal patch embeddings are averaged to produce a single fixed-dimensional embedding vector for each input patch. The pretrained model is kept frozen during both pretraining and downstream training, and embeddings are computed offline. The resulting representations are projected to the model's embedding dimension before being passed to the transformer encoder.

**Masking.** We employ a random masking strategy in which 50% of tokens are independently selected via a Bernoulli distribution and replaced with a learnable mask token. Masking is applied after temporal feature extraction, ensuring a consistent masking scheme across all methods, including the pretrained MOMENT where embeddings are computed offline.

### 3.3. Spherical Positional Encoding (SPE)

We use Spherical Positional Encoding (SPE) (Yuce & Stober, 2026) to incorporate spatial information from electrode locations, as illustrated in Fig.1. The encoding introduces no additional learnable parameters and generalizes across montages, and is kept fixed across all experiments to isolate the effect of temporal feature extractors. For each channel, azimuth angle ($\theta$) and inclination angle ($\phi$) are obtained from canonical electrode positions and mapped using multi-frequency sinusoidal functions:

$$pos_c = [\sin(\omega_i\theta), \cos(\omega_i\theta), \sin(\omega_i\phi), \cos(\omega_i\phi)]$$

For temporal structure, standard sinusoidal positional encoding (Vaswani et al., 2017) is applied over patch indices to obtain the temporal positional encoding $pos_w$. The final positional encoding is computed as $P_{cw} = pos_w + pos_c$.

### 3.4. Transformer Encoder, Reconstruction and Classification Head

Positional encodings are added to patch embeddings and passed through a transformer encoder consisting of self-attention and feed-forward layers. The resulting representations are fed into a reconstruction head implemented as a linear layer. For downstream evaluation, the reconstruction head is replaced with a single-layer classification head that flattens patch representations across channels and temporal segments and maps them to class logits.

The model is trained with a masked reconstruction objective, $\mathcal{L} = \|\hat{X}_M - \tilde{X}_M\|_2^2$, computed over masked patches, where $\tilde{X}_M$ and $\hat{X}_M$ denote the normalized target patches and their reconstructions, respectively.

## 4. Results

### 4.1. Experimental Setup

**Pretraining Dataset.** We pretrain the EEG foundation model on the Healthy Brain Network EEG (HBN-EEG) (Shirazi et al., 2024; Alexander et al., 2017; Langer et al., 2017), a large-scale collection of EEG recordings from over 3,000 participants acquired with 128 channels. We use the publicly available preprocessed version, down-sampled to 100 Hz and band-pass filtered between 0.5–50 Hz. Recordings are segmented into non-overlapping 10-second epochs before temporal patch extraction.

**PhysioNet MI.** For motor imagery classification, we use the PhysioNet EEG Motor Movement Dataset (Goldberger et al., 2000; Schalk, 2009; Schalk et al., 2004), which contains 64-channel EEG recordings from 109 healthy subjects. Signals are recorded at 160 Hz, and each trial lasts 4 seconds.

*Table 1.* Linear probe performance across datasets.

| | PhysioNet-MI | | | FACED | | |
|---|---|---|---|---|---|---|
| Model | Balanced Accuracy ↑ | Cohen's Kappa ↑ | Weighted F1 ↑ | Balanced Accuracy ↑ | Cohen's Kappa ↑ | Weighted F1 ↑ |
| Linear | **0.547 ± 0.008** | **0.397 ± 0.011** | 0.546 ± 0.008 | 0.362 ± 0.006 | 0.279 ± 0.006 | 0.358 ± 0.005 |
| Conv | 0.546 ± 0.006 | 0.394 ± 0.008 | **0.547 ± 0.007** | 0.397 ± 0.012 | 0.318 ± 0.013 | 0.391 ± 0.010 |
| MOMENT | 0.526 ± 0.008 | 0.369 ± 0.011 | 0.525 ± 0.010 | **0.398 ± 0.006** | **0.321 ± 0.007** | **0.393 ± 0.005** |

*Table 2.* Fine-tuning performance across datasets.

| | PhysioNet-MI | | | FACED | | |
|---|---|---|---|---|---|---|
| Model | Balanced Accuracy ↑ | Cohen's Kappa ↑ | Weighted F1 ↑ | Balanced Accuracy ↑ | Cohen's Kappa ↑ | Weighted F1 ↑ |
| Linear | **0.582 ± 0.016** | **0.443 ± 0.021** | **0.583 ± 0.016** | 0.430 ± 0.018 | 0.356 ± 0.019 | 0.428 ± 0.016 |
| Conv | 0.564 ± 0.009 | 0.418 ± 0.012 | 0.564 ± 0.011 | **0.454 ± 0.012** | **0.385 ± 0.013** | **0.453 ± 0.011** |
| MOMENT | 0.576 ± 0.010 | 0.435 ± 0.013 | 0.576 ± 0.010 | 0.442 ± 0.012 | 0.369 ± 0.013 | 0.438 ± 0.011 |

**FACED.** For emotion recognition, we use the Finer-grained Affective Computing EEG Dataset (FACED) (Chen et al., 2023), comprising 32-channel EEG recordings from 123 subjects. The data are recorded at 250 Hz and segmented into non-overlapping 10-second epochs before temporal patch extraction.

All datasets are resampled to a common sampling rate of 100 Hz to ensure consistency across pretraining and downstream evaluation, and are split at the subject level to reflect cross-subject generalization. The pretraining dataset contains approximately 480k training samples, while PhysioNet-MI and FACED contain 6.6k and 7.3k downstream training samples, respectively. Additional details on datasets, pre-processing, data splits, hyperparameters, and computational setup are provided in the Appendix.

### 4.2. Downstream Task Performance

We evaluate the models on two EEG classification tasks: motor imagery and emotion recognition, under linear probing and fine-tuning protocols. In the linear probing setting, pretrained components are frozen and only the classification head is trained. In the fine-tuning setting, the transformer backbone and classification head are further updated on the downstream data. Due to computational constraints, the MOMENT encoder is kept frozen during fine-tuning. To ensure a controlled comparison, the other temporal feature extractors are also kept frozen; otherwise, performance differences could reflect fine-tuning adaptation rather than the quality of the learned representations.

Tables 1 and 2 summarize downstream performance. Friedman tests indicate statistically significant differences across models for most settings, except the FACED fine-tuning protocol. However, post-hoc comparisons did not reveal significant pairwise differences after Holm-Bonferroni correction. We therefore focus on interpreting consistent trends in mean performance across settings and metrics; detailed statistical analyses are provided in the Appendix.

On PhysioNet-MI, the linear and convolutional models achieve similar mean performance across metrics in the linear probing setting, whereas the MOMENT model shows lower average performance. After fine-tuning, the linear model attains the highest average performance, followed by MOMENT. In contrast, on FACED, both the convolutional and MOMENT models achieve higher performance than the linear baseline under both protocols. All models improve after fine-tuning, with larger gains observed on FACED than PhysioNet-MI. Although direct comparison is limited by differences in experimental protocols, the obtained performance falls within the range reported by supervised EEG baselines on these datasets (Wang et al., 2025).

Overall, the pretrained TSFM achieves competitive performance but does not consistently outperform the baseline temporal feature extractors. While broader conclusions are limited by the evaluation scope, the results demonstrate that a frozen domain-general TSFM can serve as a viable temporal feature extractor within an EEG foundation model. We note that the frozen transfer setting may constrain the model's ability to capture the temporal dynamics of EEG signals.

## 5. Conclusion

This study presents a systematic comparison of temporal feature extractors in EEG foundation models and evaluates pretrained TSFMs as frozen temporal feature extractors. Across the evaluated benchmarks, no single strategy consistently outperformed the others. Simple linear projections performed competitively on the motor imagery dataset, whereas the emotion dataset benefited from richer temporal modeling. The pretrained TSFM demonstrated its potential as a frozen temporal feature extractor despite not being adapted to EEG. Future work will investigate adapting TSFMs during pretraining and evaluating them across a broader range of downstream tasks and datasets.

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

# A. Appendix

The Appendix provides supplementary details on visualization, model hyperparameters, dataset descriptions, data preprocessing, data splits, evaluation metrics, and computational environment.

## A.1. Experimental Details

### Datasets

**Pretraining Dataset.** The Healthy Brain Network EEG dataset (HBN-EEG) (Shirazi et al., 2024; Alexander et al., 2017; Langer et al., 2017) is a large-scale collection of EEG recordings from children and adolescents aged 5 to 21 years. The recordings were acquired using an EGI HydroCel Geodesic Sensor Net with 128 channels at a sampling rate of 500 Hz. Each subject completed active and passive tasks, including resting state, surround suppression, movie watching, contrast change detection, sequence learning, and symbol search.

We use the provided preprocessed version of the dataset, which is downsampled to 100 Hz and band-pass filtered between 0.5–50 Hz. The dataset is released in 11 separate releases, each containing recordings from different subjects. In our experiments, Releases 1–9 are used for pretraining the foundation model, while Release 10 is used for validation. Release 11 is held out for future experiments and is not used in this study.

Additional preprocessing was performed using MNE-Python (Gramfort et al., 2013). Each recording is divided into 10 seconds non-overlapping epochs. We remove the reference channel, detect noisy channels, and interpolate them (Bigdely-Shamlo et al., 2015; Perrin et al., 1989) after applying average referencing to the remaining channels. As the data is already band-pass filtered, no additional notch filtering (60 Hz) was applied.

### Downstream Datasets

**Physio-Net MI.** The PhysioNet EEG Motor Movement Dataset (Physio-Net MI) (Goldberger et al., 2000; Schalk, 2009; Schalk et al., 2004) contains 64-channel EEG recorded at 160 Hz from 109 healthy subjects using the BCI2000 system with electrodes placed according to the international 10–10 system. Each subject performed 14 runs: two one-minute baseline runs and three two-minute runs of each of four motor tasks (executed and imagined left/right hand and both feet/fists movements), yielding trials of 4 seconds. No other preprocessing is applied.

**FACED.** The Finer-grained Affective Computing EEG Dataset (FACED) (Chen et al., 2023) comprises 32 channel EEG recorded from 123 subjects. Data were collected in two cohorts at sampling rates of 250 Hz and 1000 Hz respectively. The subjects watched 28 video clips designed to elicit nine emotion categories; four positive (amusement, inspiration, joy, tenderness), four negative (anger, fear, disgust, sadness), and neutral. Each trial consists of the last 30 seconds of each video clip, selected to capture peak emotional response. Each recording is divided into 10 sec non-overlapping epochs. We use the published preprocessed version of the dataset, in which electrode ordering and naming are standardized across recording cohorts, and a bandpass filter of 0.05–47 Hz was applied. No other additional preprocessing is applied.

In the downstream experiments, the datasets are split at the subject level. For both datasets, 15% of subjects are used for validation, 15% for testing, and the remaining subjects for training. A fixed random permutation with seed 42 is applied to ensure consistent subject splits across all experiments.

An overview of the pretraining and downstream datasets is provided in Table 3.

*Table 3.* Summary of Datasets

| Dataset | #subjects | #channels | length of EEG | #samples (Train-Val) or (Train-Val-Test) |
|---|---|---|---|---|
| HBN-EEG (pretrain) | 2449 | 128 | 10 seconds | 482593-64991 |
| PhysioNet-MI | 109 | 64 | 4 seconds | 6629-1429-1347 |
| FACED | 123 | 32 | 10 seconds | 7308-1512-1512 |

### Downstream Evaluation Metrics

*Balanced Accuracy* is a performance metric used to evaluate models on imbalanced datasets. It computes the average recall across all classes. Its values range from 0 to 1, where 1 indicates perfect classification.

*Cohen's Kappa* is a statistical measure of agreement between two classifiers. Its values range from $-1$ to 1, where 1 indicates perfect agreement.

*Weighted F1* is an evaluation metric for multiclass classification that computes the F1 score for each class and weights them by the number of samples in each class. It ranges from 0 to 1, where 1 indicates perfect precision and recall for all classes.

**Computational Environment** Pretraining experiments are conducted on NVIDIA H100 (80GB) GPUs, while downstream experiments are run on NVIDIA GeForce RTX 2000 and RTX 2080 GPUs. Experiments are performed using Python 3.12.3 and Python 3.10.19 across different runs. Pretraining experiments require approximately 14–20 hours of training time.

### A.2. Model Setup Details

Table 4 and Table 5 summarize the pretraining and downstream hyperparameters, respectively.

*Table 4.* Hyperparameter configuration for pretraining experiments.

| Category | Hyperparameter | Value |
|---|---|---|
| Input | Sampling rate | 100 Hz |
| | Patch length | 200 samples (2 s) |
| | Input shape | $128 \times 1000$ |
| | Normalization | Absolute Maximum Normalization / RevIN (MOMENT) |
| Temporal Feature Extractor | Linear baseline | Linear projection |
| | Conv baseline | 2D CNN encoder (kernel size (1,31), (1,15), (1,7), (1,3)) |
| | TSFM | Frozen MOMENT-small |
| Masking | Masking strategy | Random token masking |
| | Mask ratio | 50% |
| | Selection rule | Bernoulli sampling |
| | Mask token | Learnable |
| Transformer | Encoder layers | 10 |
| | Hidden dimension | 192 |
| | Attention heads | 8 |
| | FFN dimension | 800 |
| | Positional encoding | Spherical PE |
| | Dropout | 0.1 |
| Training | Weight Initialization | Sinusoidal-Initialization ((Fernández-Hernández et al., 2025)) |
| | Objective | Masked reconstruction |
| | Optimizer | AdamW |
| | Learning rate scheduler | CosineAnnealingLR |
| | Learning rate | $5e-4$ |
| | Weight decay | $5e-2$ |
| | Batch size | 32 |
| | Epochs | 30 |
| | Early stopping patience | 10 |

*Table 5.* Hyperparameter configuration for downstream experiments

| Hyperparameter | Value |
|---|---|
| Random seeds | [30-35] |
| Optimizer | AdamW |
| Learning rate scheduler | CosineAnnealingLR |
| Transformer Learning rate | $1e-4$ |
| Classifier Learning rate | $5e-4$ |
| Weight decay | $1e-2$ |
| Batch size | 64 |
| Epochs | 50 |
| Label smoothing | 0.1 |
| Dropout | 0.1 |
| Early stopping patience | 10 |

## A.3. Statistical Analysis

All statistical analyses were performed at a significance level of $\alpha = 0.05$. Friedman omnibus tests were used to assess performance differences across model variants, accounting for seed-level paired observations. The results indicate significant differences for most dataset $\times$ training regime $\times$ metric combinations, excluding the fine-tuning regime on FACED. The corresponding results are shown in Table 6. Post-hoc Wilcoxon signed-rank tests were conducted for pairwise model comparison; however, no comparisons remained statistically significant after conservative Holm-Bonferroni correction. Results of post-hoc analyses for Balanced Accuracy scores, including raw p-values, are reported in Table 7.

*Table 6.* Friedman test results across model variants.

| Dataset | Setting | Metric | Friedman $\chi^2$ | $p$-value |
|---------|---------|--------|-------------------|-----------|
| PhysioNet-MI | Linear Probe | Bal. Acc. | 9.33 | 0.0094 |
| | | Cohen's Kappa | 9.00 | 0.0111 |
| | | W-F1 | 9.33 | 0.0094 |
| | Fine-Tuning | Bal. Acc. | 7.00 | 0.0302 |
| | | Cohen's Kappa | 9.33 | 0.0094 |
| | | W-F1 | 7.00 | 0.0302 |
| FACED | Linear Probe | Bal. Acc. | 9.33 | 0.0094 |
| | | Cohen's Kappa | 9.00 | 0.0111 |
| | | W-F1 | 9.00 | 0.0111 |
| | Fine-Tuning | Bal. Acc. | 3.00 | 0.2231 |
| | | Cohen's Kappa | 4.33 | 0.1146 |
| | | W-F1 | 4.33 | 0.1146 |

*Table 7.* Post-hoc Wilcoxon signed-rank tests with Holm-Bonferroni correction for Balanced Accuracy scores. Post-hoc tests are reported only for settings in which the Friedman omnibus test was significant.

| Dataset | Setting | Comparison | $p_{raw}$ | $p_{Holm}$ |
|---------|---------|------------|-----------|------------|
| PhysioNet-MI | Linear Probe | Linear vs Convolution | 0.5625 | 1.0000 |
| | | Linear vs MOMENT | 0.0313 | 0.0938 |
| | | Convolution vs MOMENT | 0.0313 | 0.0938 |
| | Fine-Tuning | Linear vs Convolution | 0.0313 | 0.0938 |
| | | Linear vs MOMENT | 0.4375 | 1.0000 |
| | | Convolution vs MOMENT | 0.0625 | 0.1875 |
| FACED | Linear Probe | Linear vs Convolution | 0.0313 | 0.0938 |
| | | Linear vs MOMENT | 0.0313 | 0.0938 |
| | | Convolution vs MOMENT | 1.0000 | 1.0000 |

## A.4. Additional Visualizations

### Training Loss Plots

The pretraining loss curves for the three variants of the EEG foundation model are shown in Fig. 2. The loss decreases gradually over epochs for all models. The loss values of the MOMENT-based model are not directly comparable to the other variants due to differences in normalization, which result in different input value ranges. Therefore, cross-model comparison is based primarily on downstream task performance, while reconstruction curves are interpreted in relative terms.

### Reconstruction Plots

We provide qualitative reconstruction examples for masked and unmasked tokens in Figure 3 to illustrate the behavior of different temporal feature extractors. The top row presents reconstructions for masked tokens, where the input is not observed by the model. The bottom row shows reconstructions for unmasked tokens. Note that the reconstruction loss is computed only over masked tokens during training. Therefore, unmasked token examples are provided for qualitative illustration only and are not used for evaluation. Across the masked examples, convolutional model better preserves local variations while MOMENT produces smoother reconstructions. These examples are based on a randomly selected sample from test set and are intended for illustrative purposes only.

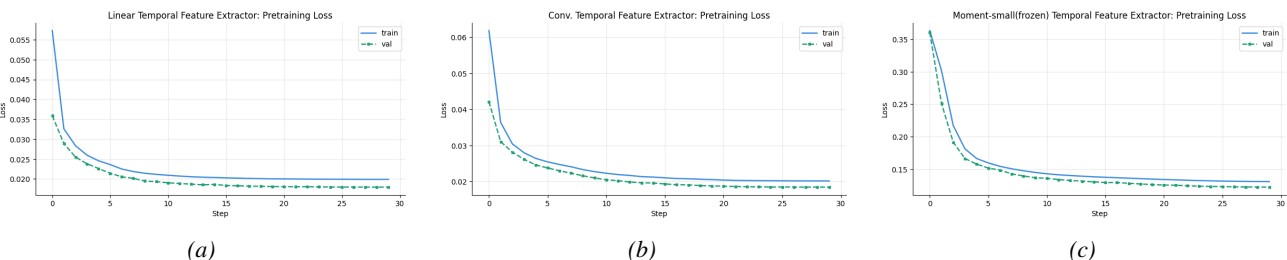

*Figure 2.* Pretraining loss curves (train and validation) across 30 epochs for three model variants: (a) Linear, (b) Conv temporal feature extractor, and (c) MOMENT-small feature extractor.

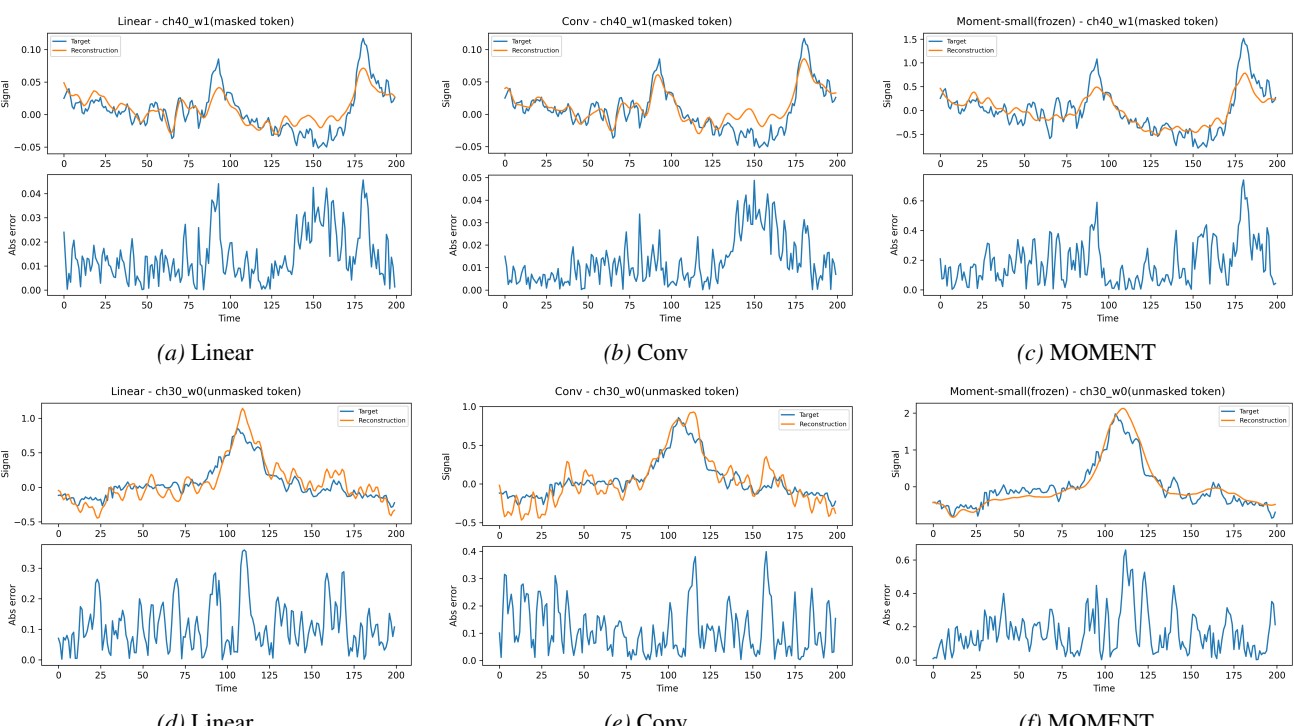

*Figure 3.* Reconstruction comparison for masked (top row) and unmasked (bottom row) tokens across three model variants: Linear, convolutional temporal feature extractor, and MOMENT-small temporal feature extractor. For each subplot, the upper plot shows the target and reconstructed signals, while the lower plot shows the absolute error between the target and reconstructed signal.

