# OpenReview forum: "Temporal Feature Extractors in EEG Foundation Models: A Controlled Comparison Including a Pretrained Time-Series Model"
_ICML.cc/2026/Workshop/FMSD — FMSD @ ICML 2026 Poster_

### Official Review · Reviewer_pX3c · 2026-05-13
**Review for "Temporal Feature Extractors in EEG Foundation Models: A Controlled Comparison Including a Pretrained Time-Series Model"**

**Rating:** 6
**Confidence:** 4

**Review:**

## Summary

This paper studies temporal feature extractors in EEG under a controlled setup. It compares linear projection, convolutional encoding, and a frozen pretrained TSFM (MOMENT) on downstream EEG tasks. The paper aims to understand whether general purpose TSFM representations can transfer to EEG analysis.

## Strengths

The controlled comparison is useful because it isolates the temporal extractor from other architectural factors.

The use of normalization and spherical positional encoding is a strength. These techniques are appropriate for EEG signals, as they help preserve important signal properties and incorporate spatial information from electrode locations.

## Areas for Improvement

A major limitation is that the TSFM is kept frozen. Models such as MOMENT or Mantis usually allow fine tuning, and several related works have shown that fine-tuned TSFMs can achieve strong or even SOTA performance in EEG tasks. Therefore, evaluating only frozen TSFM features may underestimate the true potential of TSFMs in this domain.

---

### Official Review · Reviewer_NJRD · 2026-05-13

**Rating:** 4
**Confidence:** 4

**Review:**

## Summary

This paper investigates the use of pretrained time-series foundation models (TSFMs) for EEG representation learning. Specifically, the authors study whether a pretrained TSFM can serve as an effective temporal embedder within an EEG foundation model (FM). They compare three temporal embedding strategies: linear, convolutional, and a frozen pretrained MOMENT encoder. They evaluate the resulting models on two downstream EEG tasks: motor imagery classification and emotion recognition. The proposed EEG FM is pretrained on the large-scale HBN-EEG dataset and incorporates spherical positional encoding to account for electrode geometry.

## Strengths

* The pretraining setup appears reasonable. The large HBN-EEG collection is used for pretraining, and the masked reconstruction framework is used (it reminds CBraMod).
* The incorporation of spherical positional encoding is an interesting design choice for EEG data, as it explicitly considers spatial electrode relationships.

## Areas for Improvement

* The central motivation and methodological setup remain somewhat unclear to me. While the paper is motivated by transferring pretrained TSFMs to EEG, the experiments do not directly evaluate a TSFM on EEG downstream tasks (e.g., via linear probing or full fine-tuning). Instead, the pretrained MOMENT model (6 transformer layers) is used as a frozen temporal embedder within a slightly larger EEG FM (10 transformer layers). As a result, it is difficult to assess whether the work truly demonstrates successful TSFM transfer to EEG.
* The empirical improvements obtained from MOMENT appear limited. Based on the presented results, the pretrained embedder improves performance only on the FACED dataset under linear probing, and the gains seem insignificant. This weakens the evidence supporting the proposed approach.
* The proposed EEG FM is not compared against existing EEG-specific foundation models or directly against standalone TSFMs, which makes it difficult to position the contribution relative to prior work.
* Several relevant recent works on applying TSFMs to EEG are missing from the related work section.

## Detailed Comments

* The current pipeline appears tightly coupled to the temporal embedder. As I understand it, a separate EEG FM must be pretrained for each temporal embedding strategy (linear, convolutional, MOMENT). This may limit flexibility when new and stronger temporal encoders become available. It could be interesting to explore adapter-based approaches that allow replacing the temporal embedder while keeping most of the EEG FM frozen.

* The related work section should discuss and compare against recent studies on TSFMs for EEG, including:
   * Yang et al. (2026), *Do We Need Domain-Specific Time-Series Models? Insights from EEG Classification Benchmarks*.
   * Gnassounou et al. (2025), *Leveraging Generic Time Series Foundation Models for EEG Classification*.

In particular, both these papers demonstrate that TSFMs can be directly used for EEG tasks being on par with EEG baselines.

* The authors argue that MOMENT is preferable because it is pretrained using a masked reconstruction objective aimed at representation learning, unlike forecasting-oriented TSFMs. While this reasoning is understandable, it may still be worthwhile to evaluate stronger TSFMs pretrained with alternative objectives, especially since several recent models have shown strong downstream performance in time-series tasks (e.g., Mantis for classification, and TiRex, Chronos-2, or TOTO-2 for forecasting). Even negative results would provide valuable insight into transferability across pretraining paradigms.

## Justification of Score

The paper addresses an important and relevant research direction at the intersection of foundation models and EEG learning. However, the methodology to use a TSFM as a temporal patch embedder is not really intuitive nor well-motivated, while having negative empirical results. In addition, the paper lacks comparisons against existing EEG foundation models and direct TSFM baselines. I therefore find the contribution not yet sufficiently validated in its current form.

---

### Official Review · Reviewer_c8KV · 2026-05-20
**Controlled comparison of temporal feature extractors for EEG foundation models, with useful baselines but limited generality**

**Rating:** 6
**Confidence:** 3

**Review:**

**Summary:** The paper compares linear, convolutional, and frozen MOMENT temporal feature extractors for EEG foundation models, evaluated on motor imagery and emotion recognition.

**Strengths:** The question is relevant to EEG foundation models and structured/time-series representation learning. The controlled setup and inclusion of a simple linear baseline are useful and make the comparison more informative.

**Weaknesses:** The main “task-dependent” claim is too broad for the evidence. The paper evaluates only two tasks with one dataset each, so the observed differences could be dataset-specific rather than generally task-dependent. Also, the statistical analysis reports no significant corrected pairwise differences, so the trends should be presented more cautiously.

**Suggestions:** Please soften the generality of the conclusions, explicitly state the limited task/dataset coverage, and move the key statistical caveat into the main text. It would also help to clarify that MOMENT is only evaluated in a frozen-transfer setting.

**Justification:** This is a relevant and useful workshop paper with a clean comparison, but the empirical scope is limited and the claims are somewhat stronger than the results support.